# RSIn-Dataset: An UAV-Based Insulator Detection Aerial Images Dataset and Benchmark

**Feng Shuang** [1,†], **Sheng Han** [1,†], **Yong Li** [1,2,*] and **Tongwei Lu** [1,2]

1   Guangxi Key Laboratory of Intelligent Control and Maintenance of Power Equipment, School of Electrical Engineering, Guangxi University, Nanning 530004, China
2   Hubei Key Laboratory of Intelligent Robot, Wuhan Institute of Technology, Wuhan 430205, China
\*   Correspondence: yongli@gxu.edu.cn
†   These authors contributed equally to this work and shared the co-first author.

**Abstract:** Power line inspection is an important part of the smart grid. Efficient real-time detection of power devices on the power line is a challenging problem for power line inspection. In recent years, deep learning methods have achieved remarkable results in image classification and object detection. However, in the power line inspection based on computer vision, datasets have a significant impact on deep learning. The lack of public high-quality power scene data hinders the application of deep learning. To address this problem, we built a dataset for power line inspection scenes, named RSIn-Dataset. RSIn-Dataset contains 4 categories and 1887 images, with abundant backgrounds. Then, we used mainstream object detection methods to build a benchmark, providing reference for insulator detection. In addition, to address the problem of detection inefficiency caused by large model parameters, an improved YoloV4 is proposed, named YoloV4++. It uses a lightweight network, i.e., MobileNetv1, as the backbone, and employs the depthwise separable convolution to replace the standard convolution. Meanwhile, the focal loss is implemented in the loss function to solve the impact of sample imbalance. The experimental results show the effectiveness of YoloV4++. The mAP and FPS can reach 94.24% and 53.82 FPS, respectively.

**Keywords:** power line inspection; insulator detection; deep learning; convolutional neural network





## 1. Introduction

With the continuous construction of power lines and the innovative development of power grid technology, intelligent power line inspection technology based on robots and unmanned aerial vehicles (UAVs) has been widely used. As an important infrastructure of the power grid system, insulators usually play the role of electrical insulation and line support. However, problems such as fouling and chipping easily occur due to the long-term exposure outside, threatening the security and stability of power grid operation. Therefore, regular line patrol inspection is necessary for line safety [1].

Generally, there are mainly two types of power line inspection methods, i.e., manual inspection and robot inspection. The traditional manual inspection is gradually declining because of low efficiency, low precision, and high labor consumption [2]. Additionally, when facing complex terrain, such as the power lines built in valleys, as shown in Figure 1a, manual inspection is risky and costly. In recent years, the rapid developments of UAV technology and computer vision technology have brought new opportunities for power line inspection [3,4]. Insulator detection based on UAV aerial images is of great significance for intelligent power inspection. It can greatly save manpower resources and improve the monitoring efficiency. In some areas that are difficult to reach or in poor natural environments, UAVs are used for the insulator detection task, as shown in Figure 1b. Nowadays, the most common method of UAV patrols of power lines is taking insulator images during flight before importing data to ground terminals for detection. Although the patrol efficiency has been improved compared with manual patrol, there is still space

for progress. In order to further raise the inspection efficiency of UAVs, it is necessary to enable UAVs to conduct real-time insulator detection tasks during flight tasks [5].

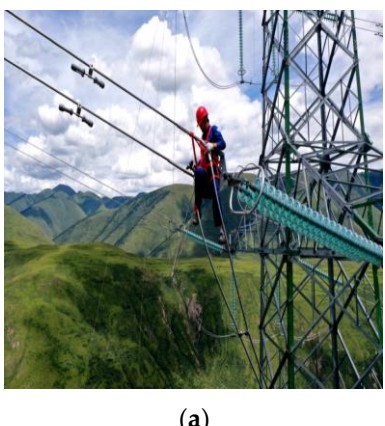 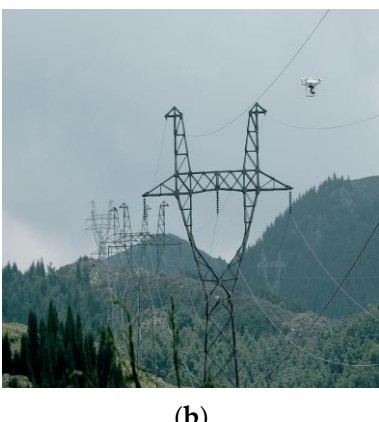

(**a**)　　　　　　　　　　　　　　　　　　　　　　　　(**b**)

**Figure 1.** Some power line inspection scenes: (**a**) manual inspection in a valley; (**b**) UAV inspection in severe weather.

At present, most object detection studies rely on large-scale and high-quality training datasets [6]. Although a variety of public datasets, e.g., Pascal VOC [7], ImageNet [8], COCO [9], ViViD++ [10], and KAIST [11], are available, there are not many line patrol inspection datasets. Generally, the insulator inspection images are obviously different from the images of the traditional datasets. The main differences are as follows: (1) the acquisition method of insulator inspection images is more difficult in real electric scenes because the power line device images can only be collected by professionals under the authorization of the power grid company. (2) The insulator inspection images have different backgrounds. Insulator datasets are specially used for power line inspection, containing more rivers, farmland, towers, houses, and other backgrounds. (3) The characteristics of the targets are diverse. Insulators show relatively different scale changes in the images taken by UAVs, and they are usually narrow and long, while the targets in traditional datasets have various shapes.

Considering the disadvantages of these public datasets and application status of deep learning methods in the insulator detection field, we summarized the existing problems into two major points [12].

- The object detection models based on pre-training on the traditional datasets are not ideal when directly applied to the insulator detection. Therefore, the insulator detection task requires new high-quality datasets for model training and testing.
- Although the existing object detection methods can be transferred to insulator detection, there is still a lack of efficient models, evaluation statistics, and benchmarks specifically for insulator object detection.

To address the above problems, we construct an insulator dataset (RSIn-Dataset) containing 1887 images and four types of insulator objects, i.e., composite insulator I, composite insulator II, glass insulator, and porcelain insulator. In addition, we make a qualitative and quantitative comparison with other datasets in terms of the number of object samples, the number of images, the number of categories, image resolution, etc. Then, we propose an insulator detection method based on YoloV4 and conduct experiments with Single Shot MultiBox Detector (SSD) [13], Faster R-CNN [14], region-based fully convolutional networks YoloV3 [15], YoloV4 [16], Yolo X [17], etc. Furthermore, an insulator detection benchmark is constructed for RSIn-Dataset.

In summary, the main contributions of this paper are as follows:

- We construct a novel dataset (RSIn-Dataset) for insulator detection in the electric power patrol scene. Compared with other datasets, RSIn-Dataset has more special power

scenarios and diversity of objectives, which can provide an important foundation for the intelligence of UAV electric power patrol based on deep learning.

- We propose the YoloV4++ network by improving YoloV4 for insulator detection. The experimental results show that YoloV4++ achieves better performance compared to other advanced networks on RSIn-Dataset.
- With the analysis of several baseline methods for object detection, the benchmark of RSIn-Dataset is constructed, which provides an important reference for future work.

## 2. Related Work

This section mainly discusses public datasets of object detection and networks for insulator detection in power line inspection. We summarized the related works from these two aspects.

### 2.1. Existing Datasets for Object Detection

The most common datasets for object detection are as follows:

- Pascal VOC Dataset: This dataset "http://host.robots.ox.ac.uk/pascal/VOC/ (accessed on 10 November 2022)" is used as a standard dataset for image detection and classification. There are two versions, i.e., voc2007 and voc2012. Voc2007 has 9963 images, while voc2012 has 17,125 images. The dataset contains 20 categories which are common in life, such as a person, bicycle, cat, bottle, etc. It has horizontal images with a size of about $500 \times 375$ pixels and a vertical image size of about $375 \times 500$ pixels. This dataset is widely used in the evaluation criteria for various object detection methods.
- Microsoft Common Objects in Context (COCO): This dataset "http://mscoco.org/ (accessed on 10 November 2022)" is a large-scale dataset available for image detection, semantic segmentation, material recognition, and image description. It has more than 330,000 images, of which 220,000 have annotated labels, containing 1.5 million targets, 80 object categories, and 91 material categories. Due to its abundance of images, deep learning methods usually carry out pre-training based on it.
- ImageNet Dataset: This dataset "https://image-net.org/ (accessed on 10 November 2022)" has more than 14 million images, covering more than 20,000 categories, of which more than 1 million images have clear categories and boundary box annotation. Deep learning methods usually choose a subset from the whole dataset for training and testing.
- Dataset for Object Detection in Aerial Images (DOTA Dataset): This dataset "https://captain-whu.github.io/DOTA/dataset.html (accessed on 10 November 2022)" is a common dataset for aerial remote sensing image object detection. There are 2806 aerial images with image resolution ranging from $800 \times 800$ to $4000 \times 4000$, containing 15 categories for a total of 188,282 instances. The images mainly contain large objects such as an airplane, ship, port, basketball court, etc. It is characterized by large changes in image spatial resolution and contains a large number of densely arranged small objects.
- Git Dataset: This dataset "https://github.com/InsulatorData/InsulatorDataSet (accessed on 10 November 2022)" is publicly available, with 848 images, divided into normal insulators (600 images) and insulators with defects (248 images). Among them, the defect insulators are synthetic images. The dataset contains only one type of insulator and insufficient kinds of power line inspection scene backgrounds. Therefore, the application scope of this dataset is limited in the insulator detection.

However, public datasets specific for insulator detection are still unavailable for the development of deep learning in the insulator detection field. Therefore, it is necessary to make a new insulator detection dataset.

*2.2. Insulator Detection Methods*

The previous power line inspection task relies on manpower. The patrol inspectors wear protective clothing and climb the insulator power tower with special equipment to diagnose the insulator. This is quite labor-consuming and inefficient [18]. With the development of the UAV, intelligent power line inspection has been rapidly promoted. UAVs are always employed to take images of power line devices. Then, the patrol inspectors diagnose the insulator through these images and raise the efficiency [19]. So, how to realize autonomous inspection of UAVs becomes the challenging problem.

In recent years, many scholars have applied the related object detection methods to the power line inspection. Wang et al. [20] designed an insulator fault detection network based on the convolutional neural network for railway insulators. The insulator detection network uses low-resolution images for position detection. The classification network uses high-resolution insulator images for fault classification. This network design improves the accuracy of insulator detection. Huang et al. [21] proposed a deep learning model based on multi-feature fusion to identify aerial insulator faults. The network integrates the manually extracted color and local binary pattern (LBP) texture features to more fully extract the effective features of the image. Sampedro et al. [22] divided the insulator string based on the fully convolutional network at first, then located and identified the insulator target. Kang et al. [23] proposed an insulator detection network based on the Faster R-CNN and multi-task neural network. This network can simultaneously perform insulator segmentation and defect detection tasks. Li et al. [24] designed a helmet detection network based on YoloV3. The k-means++ clustering algorithm is used to optimize the selection method and focal loss is introduced to reduce the weight of simple backgrounds. To reduce parameters of object detection models, Han et al. [25] improved a lightweight detection method based on YoloV4. The network uses only two feature extraction layers of the backbone for feature fusion, which improves the detection speed. Lin et al. [26] combined the Faster R-CNN module and the U-net module [27] for insulator detection. The Faster R-CNN module is used for object detection. The U-net module is used for pixel classification to achieve the localization of damaged insulators in aerial images. Considering the balance between detection speed and accuracy, Luo et al. [28] improved a method based on YoloV3, which adopts multi-scale prediction network architecture. Lin et al. [29] proposed a detection method based on the lightweight network MobileNetV1 [30] for product quality inspection.

However, most of the above proposed or improved networks improve the accuracy of the network through feature fusion or deepening the network depth, which leads to more model parameters and slow detection speed. Thus, they are not suitable for the deployment of edge devices. Some studies improve the model detection speed through the lightweight network model. However, that causes a decrease in accuracy, which cannot achieve real-time intelligent power line inspection. A few studies have aimed to achieve a balance between model speed and accuracy, but the experimental performance needs to be further improved [31].

Considering the shortcomings of the above research methods, it is challenging to propose a lightweight insulator detection method that can efficiently detect targets in real time during power line inspection.

## 3. Dataset Construction

*3.1. Insulator Image Acquisition*

RSIn-Dataset consists of three parts, i.e., Part-1, Part-2, and Part-3. For Part-1, the images are taken by the DJI M300RTK UAV work platform, with a total of 1000 images. The image size is 5472 × 3078. In the insulator image acquisition period, we use DJI M300RTK UAV to take aerial images of insulators at different places and at different times. DJI M300RTK UAV is 810 × 670 × 430 mm in size and empty weight is up to 3.6 kg. It has a maximum load of 2.7 kg and 55 min long endurance. According to the specific task, it can carry different pan, tilt, and zoom (PTZ) movements to work, as shown in Figure 2a.

Here, we choose a Zenmuse P1 camera. The Zenmuse P1 camera is $152 \times 110 \times 169$ mm in size and weighs about 930 g. The maximum resolution is up to $5472 \times 3648$. After loading the Zenmuse P1 camera, we controlled a UAV to patrol power lines to take the insulator images, as shown in Figure 2b. For Part-2, the images are from the Git dataset, with a total of 848 images. The image size is $1152 \times 864$. For Part-3, the images are 39 insulator images from the Internet, whose maximum size is up to $7360 \times 4912$. We mark the insulators in each image of our dataset.

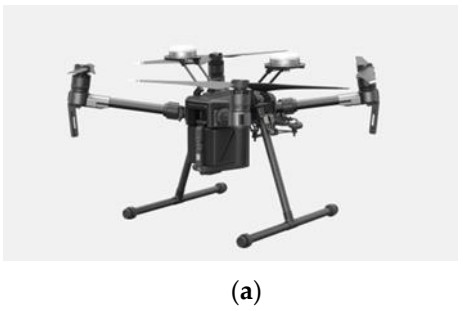 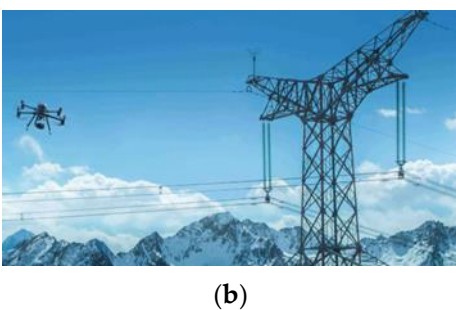

(**a**)                                                (**b**)

**Figure 2.** Data acquisition: (**a**) image acquisition vehicle; (**b**) image acquisition process. These two images were acquired from "https://www.dji.com/ (accessed on 1 January 2023)".

In total, RSIn-Dataset contains 1887 images and 3286 insulator targets, with image resolution ranging from $1152 \times 864$ to $7360 \times 4912$. RSIn-Dataset contains insulators of different sizes in the power line. We divide them into four types of insulators of different colors and shapes, i.e., composite insulator I, composite insulator II, glass insulator, and porcelain insulator, as shown in Figure 3. In addition, the backgrounds are complex and diverse, covering rivers, farmland, towers, and houses, which are very representative.

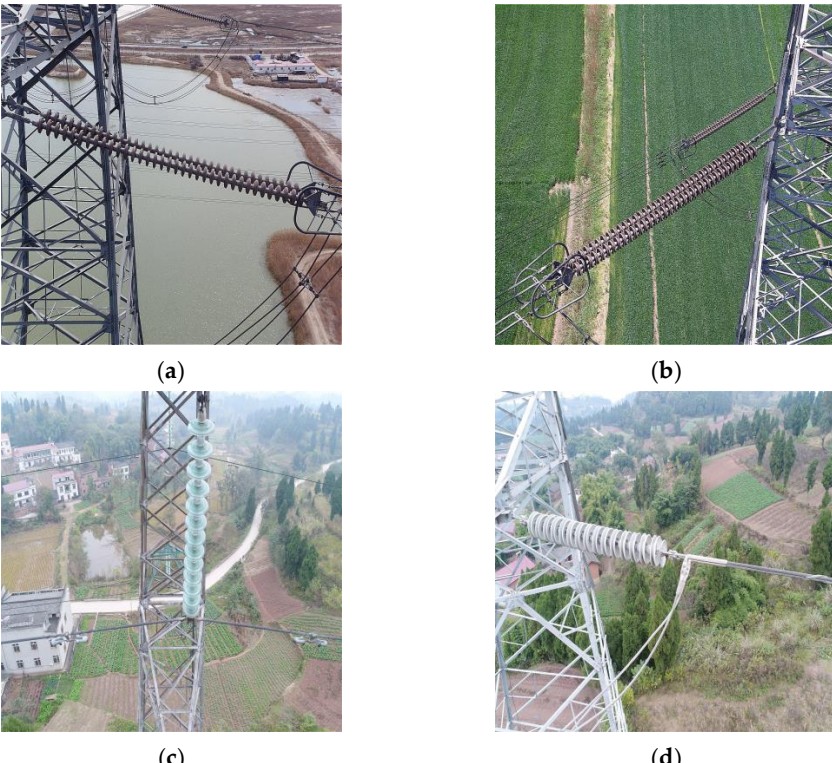

**Figure 3.** The different types of insulators: (**a**) composite insulator I; (**b**) composite insulator II; (**c**) glass insulator; (**d**) porcelain insulator.

### 3.2. Dataset Labeling

Here, we use LabelImg "https://github.com/tzutalin/labelImg (accessed on 5 July 2022)"to label the datasets. During making labels, we use rectangular bounding boxes to completely cover the object, labeled: "insulator1", "insulator2", "insulator3", and "insulator4". The four labels represent composite insulator I, composite insulator II, glass insulator, and porcelain insulator, respectively, as shown in Figure 4. Moreover, since the insulator strings in the dataset are two columns, we label them in the same rectangular box. In addition, insulator strings with a degree of being covered greater than 1/3 are not marked.

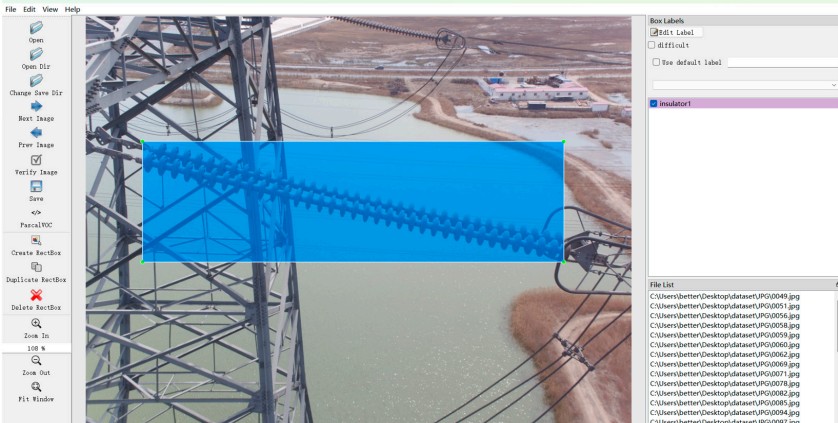

**Figure 4.** Data Labeling.

To ensure the quality of the insulator dataset, we employ 3 object detection practitioners for the data labeling and strictly perform the correct labeling procedure. In the first round of labeling, each image is manually labeled by 3 researchers, including the object category labels and the coordinates of the rectangular box. In the second round of inspection, the researchers check the data label and vote whether the image is qualified. Finally, we label a total of 3,286 object bounding boxes with four categories.

### 3.3. Dataset Statistical Analysis

RSIn-Dataset contains a large number of labeled samples. The minimum number of samples for each insulator category exceeds 300. Figure 5 shows the number and proportion of samples for each type of insulator. Specifically, 1454 samples are composite insulator I, 767 samples are composite insulator II, 736 samples are glass insulator, and 329 samples are porcelain insulator.

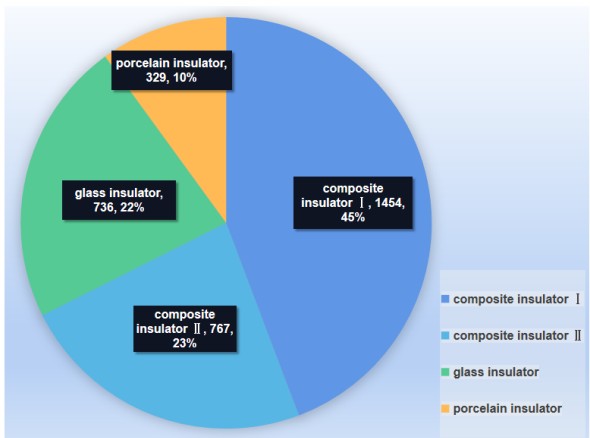

**Figure 5.** The number and proportion of each type of insulator.

To show the characteristics of RSIn-Dataset, we make a comparison with other datasets, as shown in Table 1. The public object detection datasets Pascal VOC 2007, COCO, and ImageNet basically contain no power inspection scenarios and insulator images. Actually, in object detection field, the model is usually pre-trained on the above public datasets. Compared with the DOTA dataset, although it is commonly used as a dataset for object detection of remote sensing images, it contains few insulator images. In addition, the insulator target is small in those images. Compared with the Git dataset, RSIn-Dataset contains more abundant background information. Each type of sample comes from different power line distribution scenes, including mountains, suburban fields, and river areas. These scenes are closer to the real power line inspection scenes. Additionally, RSIn-Dataset contains partial small targets and overlapping targets, which increases the difficulty of the insulator detection task. Therefore, RSIn-Dataset can help to evaluate the advantages and disadvantages of different object detection methods.

**Table 1.** Comparisons of RSIn-Dataset and other datasets.

| Dataset | Pascal VOC 2007 | COCO | ImageNet | DOTA | Git | Ours |
|---|---|---|---|---|---|---|
| Electric Scene | No | No | No | No | Yes | Yes |
| Resolution | $375 \times 500$ | / | / | $800 \times 800$ $4000 \times 4000$ | $1152 \times 864$ | $1152 \times 864$ $7360 \times 4912$ |
| Number of Categories | 20 | 91 | 20,000+ | 15 | 1 | 4 |
| Number of Images | 9963 | 330,000+ | 14,000,000+ | 2806 | 848 | 1887 |
| Number of Samples | 24,640 | 1,500,000+ | 1,000,000+ | 188,282 | 1262 | 3286 |
| Number of Insulators | Few | Few | Few | Few | 1262 | 3286 |

## 4. Baseline Methods and The Proposed YoloV4++

### 4.1. Baseline Methods

To better evaluate the effect of different object detection methods and provide a reference for power device target detection community, we introduce the baseline methods, i.e., SSD, Faster R-CNN, YoloV3, YoloV4, and Yolo X, to construct the benchmark.

The SSD algorithm mainly has three steps [13]. The SSD algorithm sends an image to the backbone network to first extract effective feature maps. Then, six appropriate feature maps from the backbone are selected. Different scale bounding boxes are set on the six selected feature maps. The SSD algorithm performs category prediction and position regression tasks on the bounding boxes respectively. Finally, the detection boxes are generated after using non-maximum suppression (NMS) strategy to screen the bounding boxes.

The Faster R-CNN algorithm is mainly divided into four steps [14]. The image is sent to the backbone network to first extract an effective feature map. Then, the proposals are extracted by the region proposal network (RPN) module from the feature map. Next, the feature map from the backbone network and proposal information from the RPN module are fused in the region of interest (ROI) pooling layer. Different scales of proposal feature maps are generated. Finally, Faster R-CNN obtains the detection boxes after calculating the feature vector from the ROI pooling.

YoloV3, YoloV4, and Yolo X have three main steps [15–17]. In the first two steps, these algorithms first obtain effective feature maps from the backbone network. Then, three feature maps are selected to build the feature pyramid [32]. The three selected feature maps are respectively located in the shallow, middle, and deep layers of the backbone network, which can help to realize the feature fusion. The feature pyramid outputs three different scale feature maps, which undertake predicting objects of different scales. In the third step, YoloV3 and YoloV4 realize both category prediction and position regression tasks simultaneously with a convolutional branch [33,34]. However, category prediction and position regression are implemented with two separate convolutional branches in Yolo X [35,36].

For the above detectors, we use stochastic gradient descent (SGD) as a backpropagation algorithm and progressively reduce the learning rate. Considering the network depth and other factors of each detector, we appropriately set iterative steps and initial learning rates to ensure the convergence of the network. For relatively deep networks, a small initial learning rate is set to avoid gradient bursting. Each detection algorithm is trained for 100 epochs, with the initial learning rate set to $1 \times 10^{-3}$ and $1 \times 10^{-4}$, respectively. Then, the learning rate drops to a tenth from the 51st epoch. The hyperparameters of the 5 detectors are shown in Table 2.

**Table 2.** Hyperparameters in the training.

| Hyperparameters | SSD | Faster R-CNN | YoloV3 | YoloV4 | Yolo X |
|---|---|---|---|---|---|
| Epoch | 100 | 100 | 100 | 100 | 100 |
| Initial learning rate | 0.001 | 0.0001 | 0.001 | 0.0001 | 0.001 |
| Batch size | 4 | 4 | 4 | 4 | 4 |
| Momentum | 0.9 | 0.9 | 0.9 | 0.9 | 0.9 |
| IoU threshold | 0.5 | 0.5 | 0.5 | 0.5 | 0.5 |

*4.2. The Proposed Method*

The flowchart of our work is shown in Figure 6. It contains two parts: dataset construction (Figure 6a) and detection network (Figure 6b). In the dataset construction part, UAV-based insulator images of power line inspection and images from the Git dataset and the Internet are collected. After labeling these images, RSIn-Dataset is constructed. Then, the detection network is trained and tested based on RSIn-Dataset. The insulator images are sent to YoloV4++ to extract feature maps. After calculating the feature vector, the final category and position are outputted from YoloV4++.

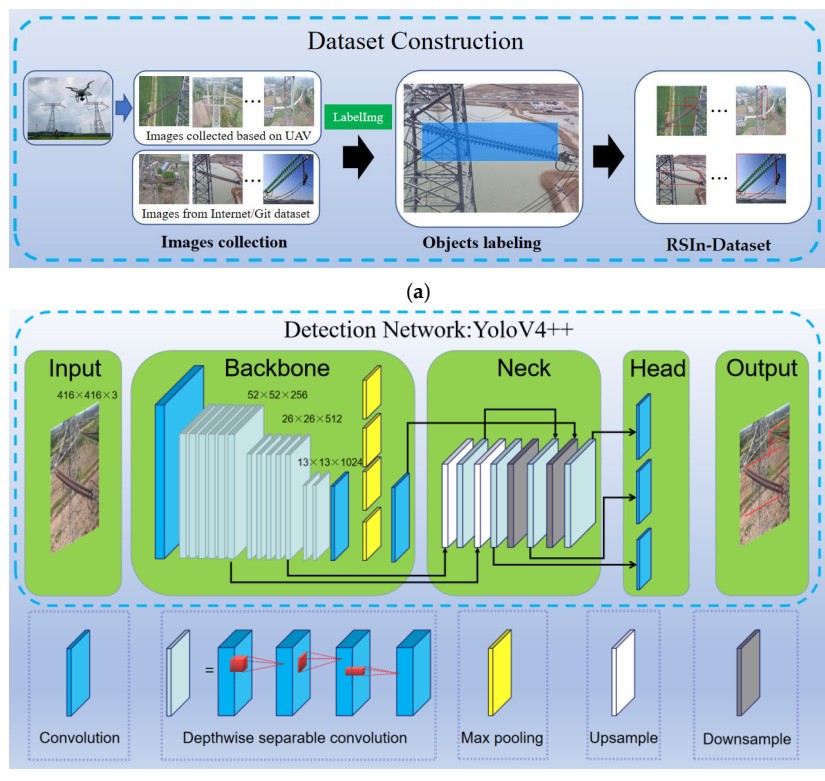

**Figure 6.** Pipeline of our work:(**a**) dataset construction; (**b**) the framework of YoloV4++.

### 4.2.1. The Structure of the Proposed YoloV4++

YoloV4++ is a lightweight network algorithm based on YoloV4. We adopt MobileNetv1 as the backbone of YoloV4++, and the depthwise separable convolution is employed in the subsequent $3 \times 3$ and $5 \times 5$ standard convolution. Additionally, the focal loss function is used for the network loss calculation. The improved object detection method is named YoloV4++, whose framework is shown in Figure 6b.

YoloV4 uses the Cross Stage Partial Dark Network (CSPDarkNet) as the backbone network. The CSPDarkNet extracts effective feature maps through the stack of residual block modules. It is the standard convolution in these residual block modules that brings huge model parameters. As we know, MobileNetv1 [30] is a lightweight neural network. It is based on the depthwise separable convolution that decomposes the standard convolution into a depthwise convolution and a pointwise convolution with a kernel of $1 \times 1$, as shown in Figure 7.

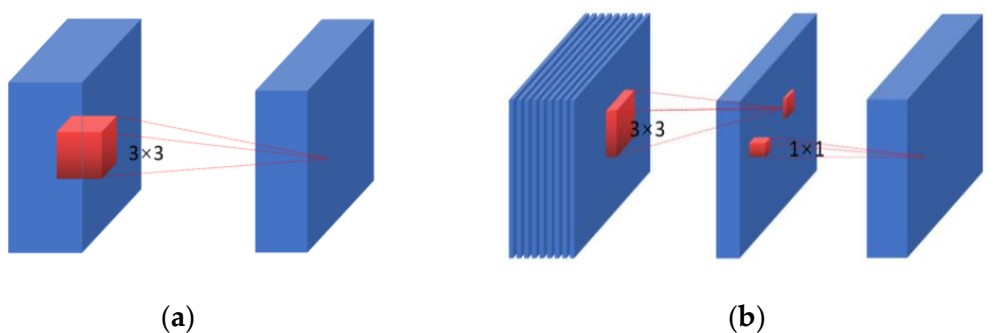

(a)          (b)

**Figure 7.** Evolution of separable convolution blocks: (**a**) standard convolution; (**b**) depthwise and pointwise convolution.

The computational cost of the depthwise separable convolution is much lower than that of the standard convolution. One convolutional layer is parameterized by its input channel $M$, output channel $N$, kernel size $D_k \times D_k$, and input layer size $D_F \times D_F$.

Standard convolution multiplies the corresponding values of the convolution filters and the input image, then sums them, as shown in Figure 8a. The calculation of standard convolution $C_1$ is:

$$C_1 = M \times D_k \times D_k \times N \times D_F \times D_F. \tag{1}$$

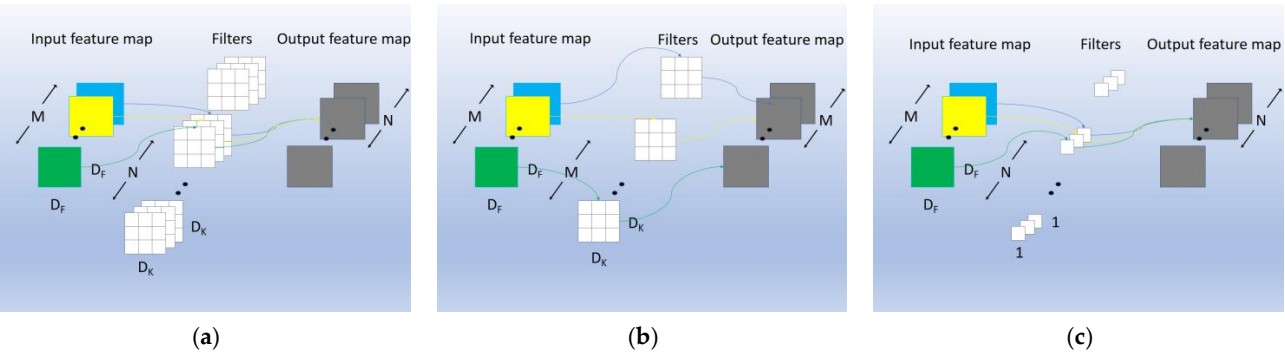

(a)          (b)          (c)

**Figure 8.** Different convolution: (**a**) standard convolution; (**b**) depthwise convolution; (**c**) $1 \times 1$ convolution called pointwise convolution in depthwise separable convolution.

Depthwise convolution is applied to each channel of the input image, as shown in Figure 8b. The pointwise convolution combines the channel output with a convolution

kernel size of $1 \times 1$, as shown in Figure 8c. The calculation of depthwise separable convolution $C_2$ is:

$$C_2 = M \times D_k \times D_k \times D_F \times D_F + M \times N \times D_F \times D_F. \tag{2}$$

Through (1) and (2), we can compare the calculated amount between depthwise separable convolution and standard convolution:

$$\frac{c_2}{c_1} = \frac{M \times D_K \times D_K \times D_F \times D_F + M \times N \times D_F \times D_F}{M \times Dk \times Dk \times N \times DF \times DF} = \frac{1}{N} + \frac{1}{D_k^2} \tag{3}$$

Generally, the output channel N is relatively large and the kernel size $D_k$ is set to 3. Therefore, we can know that the network constructed by depthwise separable convolution is about $1/9$ that by the standard convolution, which greatly reduces the model parameters.

### 4.2.2. Focal Loss for Insulator Detection

YoloV4 uses multi-scale features to detect targets of different sizes. However, most of the detection boxes contain no target, so they are labeled as negative samples in training. The imbalance in the number of positive and negative samples causes negative samples to dominate the direction of the gradient update, limiting the network learning ability.

Additionally, YoloV4 calculates the category confidence loss based on the cross entropy function. The cross entropy function is shown in Equation (4):

$$CE(P_t) = -\log P_t, \tag{4}$$

where CE is the cross entropy loss and $P_t$ is the probability of positive and negative samples. The total loss is added with the same weight for all samples, whether it is easy or difficult to classify. In the case of an imbalance of positive and negative samples, the training model will be overfitting for the negative samples and the positive samples cannot be effectively trained.

To address these two problems, we introduce the focal loss to optimize the loss function of YoloV4. The loss function of YoloV4 is composed of a multi-part loss-weighted sum, as shown in Equations (5)–(8). The specific calculation equations refer to [14]:

$$Loss = loss\_loc + loss\_conf + loss\_cls, \tag{5}$$

$$\begin{aligned} loss\_loc = \omega_{coord} \times \sum_{i=0}^{S \times S} \sum_{j=0}^{B} 1_{i,j}^{obj} \\ \times (2 - w_i \times h_i) \left[ (x_i - x_i')^2 + (y_i - y_i')^2 + (w_i - w_i')^2 + (h_i - h_i')^2 \right], \end{aligned} \tag{6}$$

$$loss\_conf = \sum_{i=0}^{S \times S} \sum_{j=0}^{B} 1_{i,j}^{obj} \times (-\log P_t) + \omega_{noobj} \times \sum_{i=0}^{S \times S} \sum_{j=0}^{B} 1_{i,j}^{noobj} \times (-\log P_t), \tag{7}$$

$$loss\_cls = \omega_{cls} \times \sum_{i=0}^{S \times S} \sum_{j=0}^{B} \sum_{1}^{n} 1_{i,j}^{obj} \times t_i (-\log t_i'), \tag{8}$$

where *Loss* is the total loss of YoloV4, *loss_loc* is box regression loss, *loss_conf* is confidence loss, and *loss_cls* is category loss. $\omega_{noobj}$, $\omega_{coord}$, and $\omega_{cls}$ are the weight coefficients. S is the grid size, B is the number of bounding boxes, i and j are the horizontal and vertical coordinates of the grid. $1_{i,j}^{obj}$ and $1_{i,j}^{noobj}$ refer to 1 and 0 when the (i, j) grid has an object. $x_i$, $y_i$, $w_i$, $h_i$ are the prediction values of detection boxes. $x_i'$, $y_i'$, $w_i'$, $h_i'$ are the true values of detection boxes. n is the number of categories. $t_i$ is the true value of each category and $t_i'$ is the prediction value of each category.

The contribution of the positive and negative sample is in confidence loss of the total loss results, we employ the focal loss [37] in this part. The focal loss function is shown in Equation (9):

$$FL = -\alpha_t 1 - P_t{}^\gamma \log P_t. \tag{9}$$

In Equation (9), $\alpha_t$ is the equilibrium parameter, $\gamma$ is the modulation coefficient, and $P_t$ is the probability of positive and negative samples. By adjusting the parameter $\alpha_t$, we can control the contribution of the positive and negative samples to the confidence loss. In addition, the contribution proportion of difficult or easy samples in the confidence loss can be controlled by the parameter $\gamma$. When $\gamma > 0$, the loss of easily classified samples will be reduced while the counterpart of hard to classify samples will increase. Thus, the learning ability of the model for samples can be enhanced. We set $\alpha_t$ to 0.25 and $\gamma$ to 2 during the experiment. Total YoloV4++ algorithm loss and *Focal loss_conf* are shown in Equations (10) and (11):

$$Loss\_new = loss\_loc + Focal\ loss\_conf + loss\_cls. \tag{10}$$

$$
\begin{aligned}
Focal\ loss\_conf = {} & \sum_{i=0}^{S \times S} \sum_{j=0}^{B} 1_{i,j}^{obj} \times (-\log P_t) \times \alpha_t (1 - P_t)^\gamma \\
& + \omega_{noobj} \times \sum_{i=0}^{S \times S} \sum_{j=0}^{B} 1_{i,j}^{noobj} \times (-\log P_t) \times \alpha_t (1 - P_t)^\gamma
\end{aligned}
\tag{11}
$$

where *Loss _new* is the total loss of YoloV4++, *loss_loc* is box regression loss, focal *loss_conf* is confidence loss based on focal loss, and *loss_cls* is category loss.

### 4.2.3. Implementation and Evaluation Metrics

To test the YoloV4++ and build a benchmark for RSIn-Dataset, we train and test mainstream algorithms (SSD, Faster R-CNN, YoloV3, YoloV4, and Yolo X) on RSIn-Dataset. Using the 1,887 images in the dataset, the training set, validation set, and test set are randomly divided in an 8:1:1 ratio. All detector models are pre-trained on Pascal VOC 2007. Additionally, the training and testing images are resized to a fixed size of $300 \times 300$ pixels for SSD, $600 \times 600$ pixels for Faster R-CNN, $416 \times 416$ pixels for YoloV3 and YoloV4, and $640 \times 640$ pixels for Yolo X.

The experimental hyperparameters of YoloV4++ are as follows. It is trained for 100 epochs on the dataset with an initial learning of $1 \times 10^{-2}$. Then, the learning rate decreases to 1% of the original from the 51st epoch. Batch size is set to 4. Momentum and intersection of union (IoU) threshold are set to 0.9 and 0.5, respectively. The hyperparameters of the other algorithms are shown in Table 2.

All evaluations are carried out on Intel Core i7-3930 k (3.80 GHz) CPU (24 GB memory). All algorithms are implemented under the Ubuntu operating system based on the PyTorch framework.

Recently, the values of average precision (AP) and mean average precision (mAP) have often been used to evaluate the performance of the object detection methods. Therefore, to compare the performance of the mainstream object detection methods on RSIn-Dataset, we use AP and mAP to evaluate the detection results for each category of the above models separately.

The average precision (AP) is calculated through precision (P) and recall (R), as shown in Equations (12)–(14). Mean average precision (mAP) is obtained according to the AP of each category, as shown in Equation (15):

$$P = \frac{TP}{TP + FP}, \tag{12}$$

$$R = \frac{TP}{TP + FN}, \tag{13}$$

$$AP = \int_0^1 P(R)dR, \tag{14}$$

$$mAP = \frac{\sum AP}{N(class)}. \tag{15}$$

where TP is the number of samples which are detected consistently with the real category. FP is the number of samples which are detected inconsistently with the real category. FN is the number of samples which are mis-detected. N (class) is the number of sample categories.

We used two metrics in the next evaluation, i.e., mAP and COCO mAP. The mAP is calculated at IoU of 0.5, while COCO mAP is averaged over multiple IoU values, i.e., ten IoU thresholds from 0.5 to 0.95 with equal gap of 0.05.

The detection speed is also selected as an evaluation standard of the model performance. The larger the number, the faster the detection speed. The calculation equation is as shown in Equation (16):

$$FPS = \frac{Num(image)}{Time}, \tag{16}$$

where Num (image) is the number of detected images and Time is the time cost of the detection process.

## 5. Results and Discussion

### 5.1. Ablation Studies

We studied the improvement strategies in YoloV4++ to prove the effectiveness of these strategies, as shown in Table 3. From Table 3, we can make the following conclusions.

**Table 3.** Ablation studies on RSIn-Dataset.

| Method | Focal loss | MobileNetv1 | COCO mAP (%) | Param (MB) | FPS |
|--------|-----------|-------------|--------------|------------|-----|
| YoloV4 | No | No | 50.56 | 245.53 | 17.01 |
| YoloV4+ | No | Yes | 48.42 | 48.42 | 54.74 |
| YoloV4++ | Yes | YeS | 55.64 | 48.81 | 53.82 |

1. YoloV4+ employs MobileNetv1 as the backbone network, which brings a reduction in model parameter size. The model size of YoloV4+ is about 1/5 that of YoloV4. For detection accuracy, compared with YoloV4, YoloV4+ decreased COCO mAP by about 1.5%, while YoloV4+ achieves a huge improvement in the FPS from 17.01 to 54.74. According to the comparison of YoloV4 and YoloV4+, we can see that using MobileNetv1 as the backbone network can greatly improve the efficiency performance. It proves that the lightweight backbone network can effectively improve the model processing efficiency and significantly reduce the model size with a good accuracy for insulator detection.
2. Compared with YoloV4+, YoloV4++ introduces the focal loss to alleviate the problem of positive and negative sample imbalance, increasing COCO mAP by 6.54% and model size by 0.39MB. In addition, YoloV4++ has FPS only 0.92 less than YoloV4+. Considering the efficiency, accuracy, and model size, this strategy is effective, which has an obvious accuracy improvement and a very small reduction in efficiency for insulator detection.

According to the $AP_i$ metric, we can see the difference between the mean precisions of each category. YoloV3, YoloV4, and YoloV4 + algorithms perform well on the composite insulator I category, as shown in Table 4. This phenomenon benefits from the composite insulator I group having more samples. YoloV4, YoloV4+ and YoloV4++ calculate the total loss based on the cross entropy function, which causes the model learning more biased to the category that has more samples.

After introducing the focal loss, the $ap_i$ of each category of insulator decreases. We can see this change more intuitively in Figure 9. The mean precision of each insulator is closer to the mAP. This variation indicates that YoloV4++ can more reasonably learn from different insulator samples.

**Table 4.** The difference between AP$_i$ and mAP of insulators for different methods. The **ap$_i$** (i = 1, 2, 3, 4) represents the difference between AP$_i$ and mAP, where AP$_i$ (i = 1, 2, 3, 4) refers to the average precision of composite insulator I, composite insulator II, the glass insulator, and the porcelain insulator, respectively.

| Method | mAP (%) | ap$_1$ | ap$_2$ | ap$_3$ | ap$_4$ |
|---|---|---|---|---|---|
| YoloV3 | 75.52 | +4.29 | −6.93 | +2.18 | +0.44 |
| YoloV4 | 91.93 | +4.38 | −2.50 | +2.72 | −4.21 |
| YoloV4+ | 90.24 | +5.64 | −1.30 | +3.54 | −7.89 |
| YoloV4++ | 94.24 | +1.08 | −3.03 | −0.03 | +1.98 |

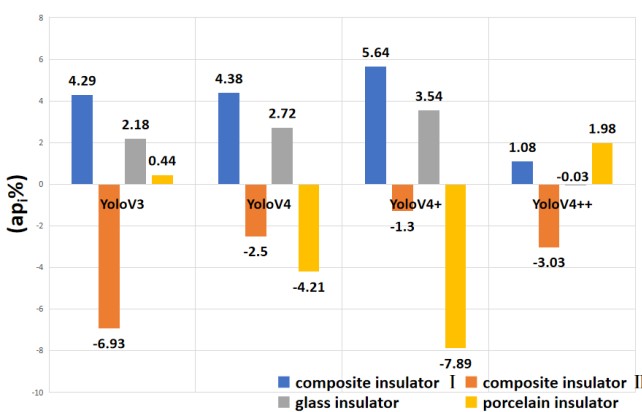

**Figure 9.** The **ap$_i$** (the difference between APi and mAP) of insulators for different methods.

*5.2. Results and Benchmark*

5.2.1. Qualitative Evaluation

In order to carry out qualitative analysis of the performance of different detectors, we show the object detection results of five baseline methods in four categories of insulator detection scenes. From the comparison in Figure 10, we can know that complex backgrounds will lead to decreasing performances of insulator detection. The interference of the background is mainly due to the existence of a large amount of noise, which causes network to struggle to fully extract the effective information of the image. Thus, the detection abilities of different methods can be compared and evaluated.

1. In ordinary scenes without complicated backgrounds, such as Figure 10a–d, SSD, Faster R-CNN, YoloV3, and YoloV4 can identify insulator targets. However, the category confidence of each target is generally low. This phenomenon indicates that the identification ability of these models is not good enough, and they easily make mistakes when working with a complex background, such as in Figure 10d,f. Yolo X and our algorithm show a very good recognition ability in this simple scenario. They can complete the detection task with no error detection or mis-detection situation. Additionally, the category confidence is generally high, close to 1.

2. In the scenario with dense insulator targets, as shown in Figure 10d, some baseline methods miss real insulator targets when detecting. In Figure 10d, the SSD algorithm misses the insulator on the lower right side of the image. Faster R-CNN, YoloV3, and YoloV4 performed poorly. The category confidence of some insulator targets is just over the threshold. In this scene, Yolo X and our algorithm still perform very well. They maintain a high category confidence in identifying the insulator target with no mis-detection or error detection.

3. With dense insulator targets against a complex background, the detection effect of SSD, Faster R-CNN, YoloV3, and YoloV4 algorithms is worse. As shown in Figure 10f, in the case where the insulators are dense and mutually masked, the YoloV4 algorithm

misses the insulator target. Although YoloV4 has no mis-detection in the scenario as in Figure 10d. This reflects that dense target detection and how to correct the background and foreground are still the goals that need to be pursued. Our algorithm adapts well, maintaining high category confidence to detect each insulator target with no error detection.

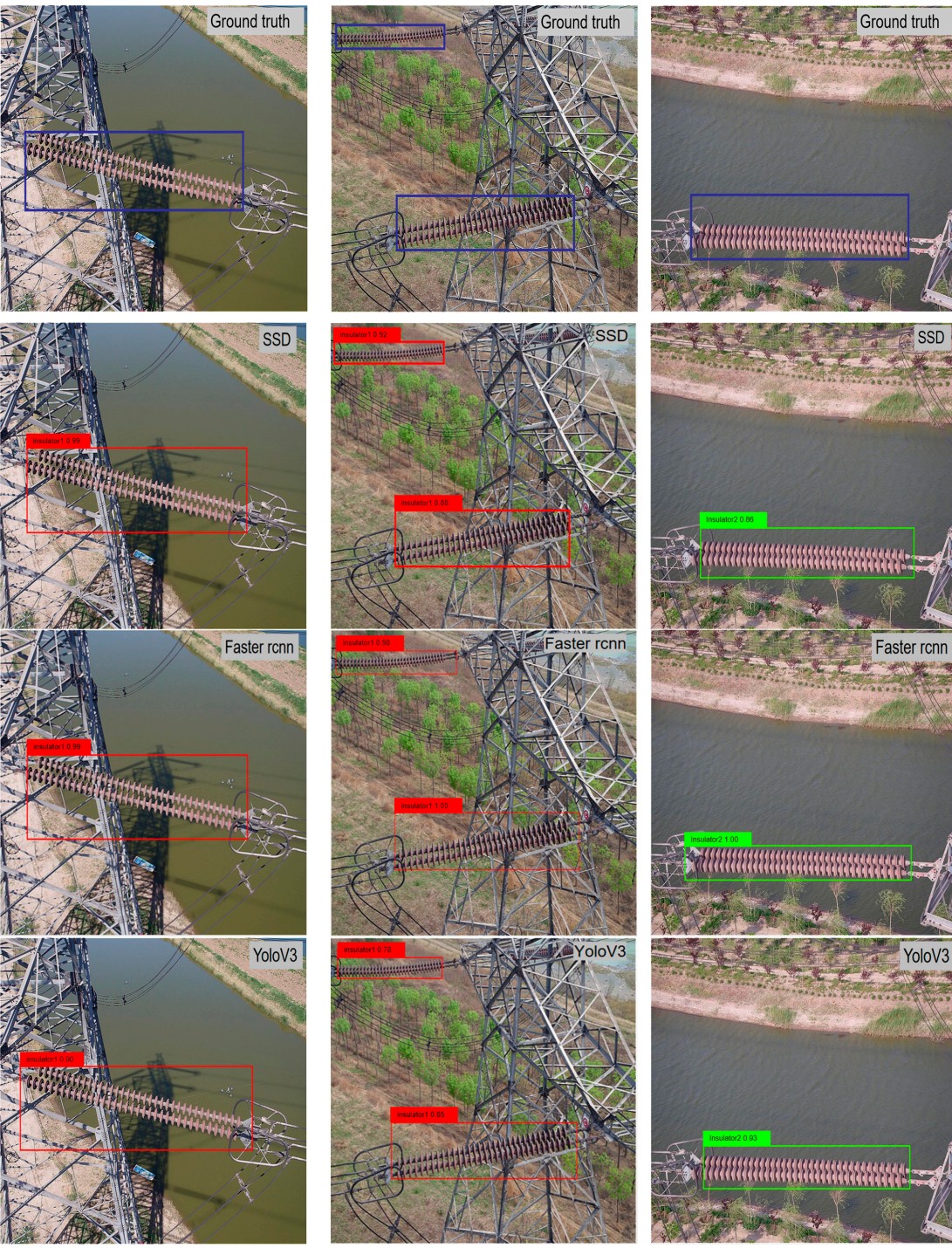

**Figure 10.** *Cont.*

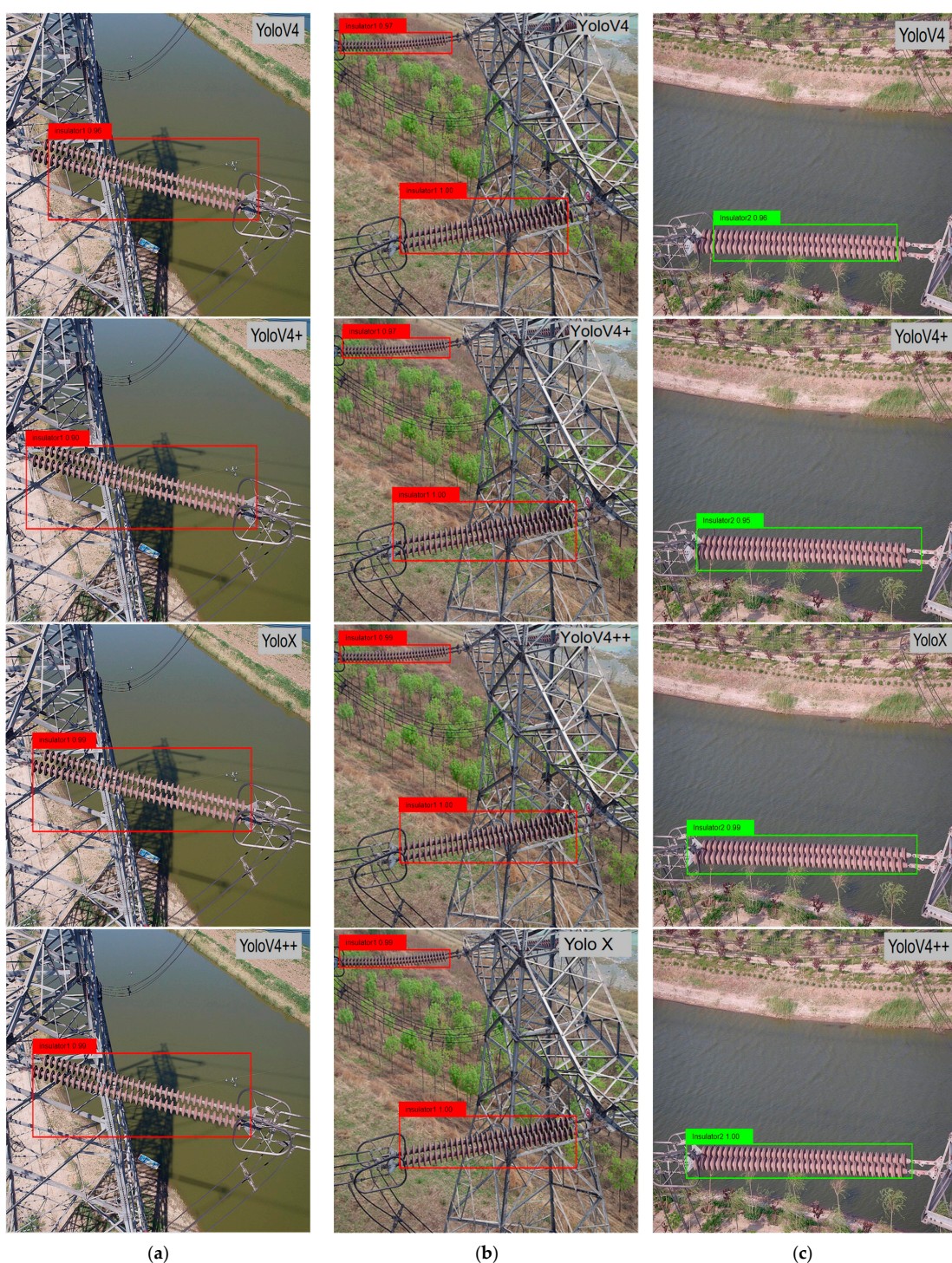

(**a**)　　　　　　　　(**b**)　　　　　　　　(**c**)

**Figure 10.** *Cont.*

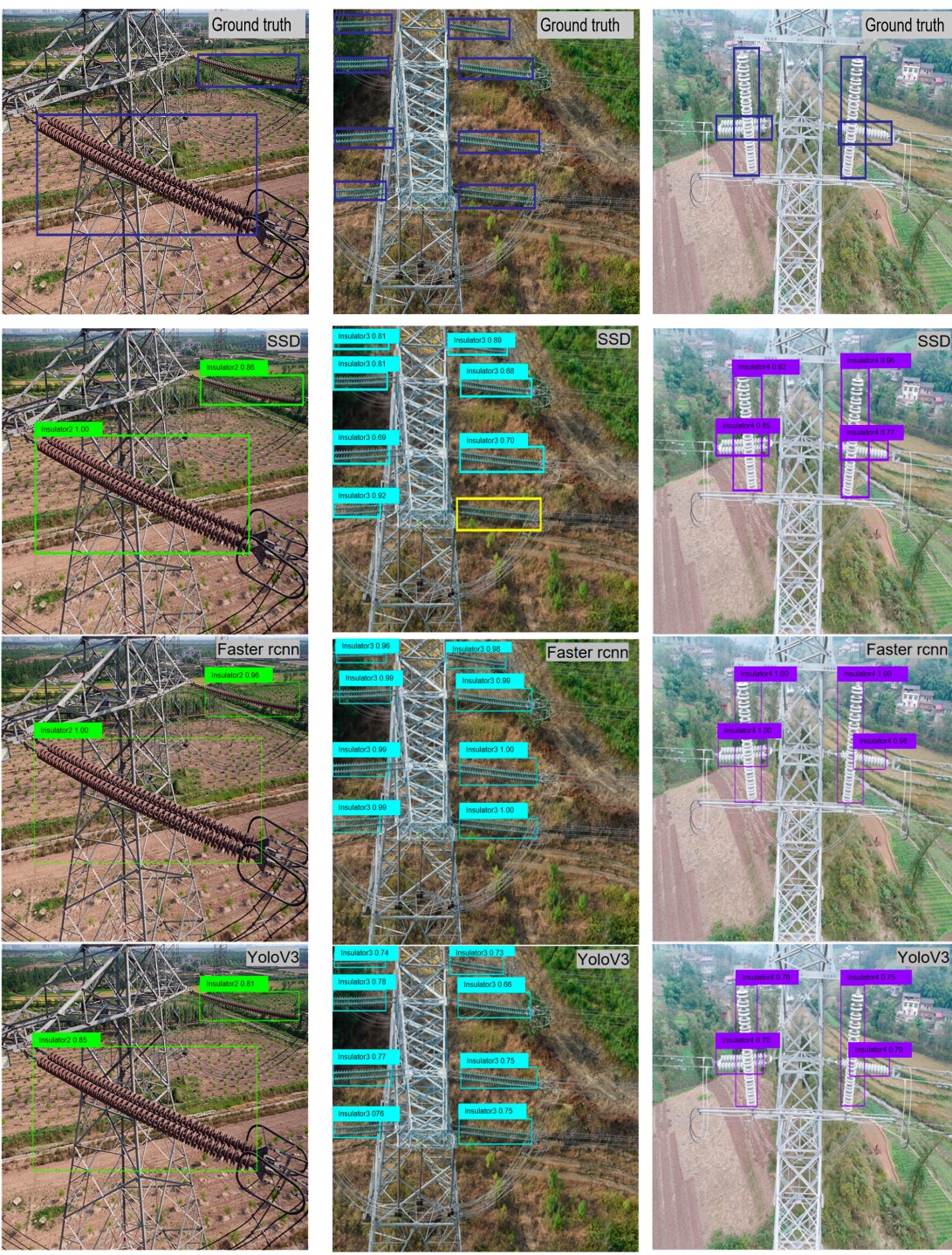

**Figure 10.** *Cont.*

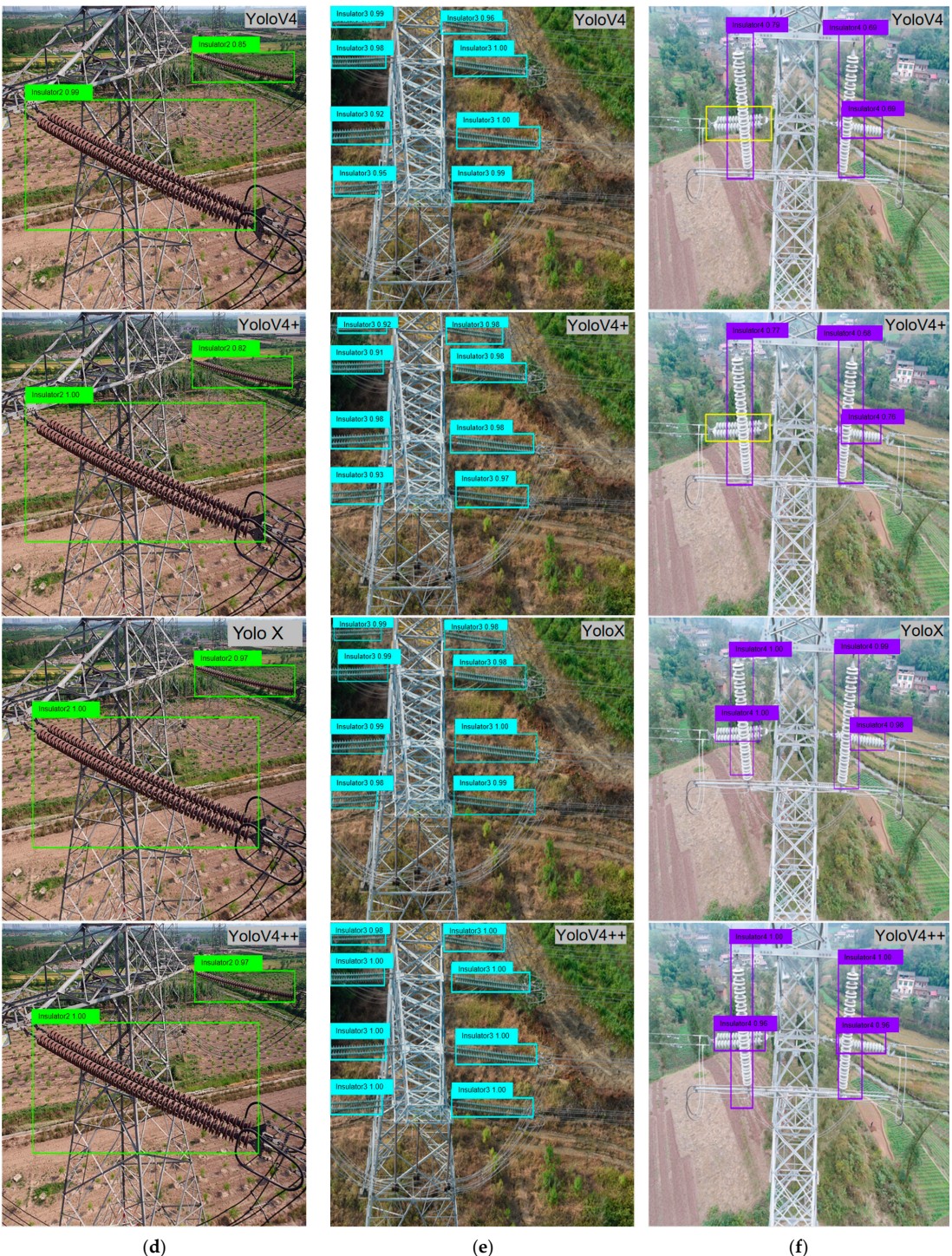

**Figure 10.** Detection results of each type of insulator against different backgrounds: (**a**) single composite insulator I; (**b**) multiple composite insulator I; (**c**) single composite insulator II; (**d**) multiple composite insulator II; (**e**) glass insulator; (**f**) porcelain insulator. The text in the upper right corner of each image represents the algorithm adopted. Red, green, cyan, and purple boxes represent composite insulator I, composite insulator II, glass insulator, and porcelain insulator. Blue boxes represent the ground truth of the insulators. Yellow boxes represent the missed insulators.

From the category confidence of each insulator, we can also know that small insulator targets acquire lower category confidence. Insulator targets which occupy more space in the image always have clearer information about color and shape than small insulator targets.

So, insulator detection networks can extract effective features and identify them well. In addition, in Figure 10b,d,e, we can see the phenomenon more clearly for the multiple insulator scenes. The category confidence of small insulator targets is always lower than close views of insulators. Although our method detects each insulator target with no mis-detection, the category confidence of small insulators is lower. So, it is necessary to pay attention to the detection of small objects for handling some extreme situations.

5.2.2. Quantitative Evaluation

In order to carry out qualitative analysis of the performance of different detectors, we evaluate and compare the performance of five detectors by AP, mAP, and COCO mAP metrics. Additionally, we list the parameter sizes of each detector model and calculate the number of pictures detected per second of each model. Table 5 shows the specific performance statistics of different detectors, where $AP_i$ (i = 1, 2, 3, 4) refers to the average precision of composite insulator I, composite insulator II, the glass insulator, and the porcelain insulator, respectively.

**Table 5.** Performance of object detection algorithms.

| Method | COCO mAP (%) | mAP (%) | $AP_1$ | $AP_2$ | $AP_3$ | $AP_4$ | Param (MB) | FPS |
|---|---|---|---|---|---|---|---|---|
| SSD | 44.54 | 84.98 | 87.80 | 80.77 | 80.25 | 91.07 | 99.7 | 35.41 |
| Faster R-CNN | 54.58 | 92.72 | 90.97 | 88.42 | 95.42 | 96.05 | 522.91 | 4.56 |
| YoloV3 | 42.52 | 75.52 | 79.81 | 68.59 | 77.70 | 75.96 | 236.32 | 22.04 |
| YoloV4 | 50.56 | 91.93 | 96.31 | 89.43 | 94.65 | 87.72 | 245.53 | 17.01 |
| YoloV4+ | 49.10 | 90.24 | 95.88 | 88.94 | 93.78 | 82.35 | 48.42 | 54.74 |
| Yolo X | 56.51 | 93.33 | 95.37 | 92.34 | 95.22 | 90.40 | 34.21 | 38.46 |
| YoloV4++ | 55.64 | 94.24 | 95.32 | 91.21 | 94.21 | 96.22 | 48.81 | 53.82 |

As shown in Table 5, the mAP values of the seven networks are: SSD 84.98%, Faster R-CNN 92.72%, YoloV3 75.52%, YoloV4 91.93%, YoloV4+ 90.24%, Yolo X 93.33%, and YoloV4++ 94.24%. The FPS values of the seven networks are: SSD 35.41, Faster R-CNN 4.56, YoloV3 22.04, YoloV4 17.01, YoloV4+ 54.74, Yolo X 38.46, and YoloV4++ 53.82. Among them, Yolo X has the best performances in COCO mAP and Param. YoloV4+ is the best in FPS. However, our method remains ahead on mAP, and FPS is 53.82, ranked second. Thus, as YoloV4++ achieves a good trade-off between mAP and FPS, our proposed network may be more advantageous than the compared networks (SSD, Faster R-CNN, YoloV3, YoloV4, and Yolo X).

The SSD algorithm is a one-stage network structure. It evenly assigns default boxes of different scales on feature maps. Although it shows a balance in detection accuracy and speed, with mAP of 84.98% and FPS of 35.41, there is still a way to go to achieve high-precision real-time detection. Faster R-CNN is designed based on a two-stage detection network, which helps raise the detection accuracy. However, this kind of design brings huge parameters. More model parameters can cause the detection speed to be far slower than that of one-stage algorithms. Here, Faster R-CNN performs the worst in detection speed, with FPS of only 4.56. The Yolo series algorithms output the object category and location at the same time, gaining advantages in speed. For example, YoloV3 is improved by about five times compared to Faster R-CNN in FPS. YoloV4 performs well in detection accuracy. However, the model parameter size of YoloV4 is still relatively large, which causes a slow speed of performance with FPS of only 17.01. To address this problem, YoloV4+ takes MobileNetv1 as its backbone, and relatively reduces the parameters, ranked first in FPS. Yolo X adopts the simple optimal transport assignment (SOTA) strategy to address the problem of sample imbalance. In addition, in the head network, it decouples the feature map to obtain the class and position information separately. The decoupling strategy makes utilization of the feature map more reasonable. Compared to SSD, Faster

R-CNN, YoloV3, and YoloV4, Yolo X achieves the best insulator detection accuracy and speed. These experimental results are consistent with their performance in public datasets, such as Pascal VOC, ImageNet, COCO, ViViD++, and KAIST [13–17].

YoloV4++ uses cross stage partial (CSP) and spatial pyramid pooling (SPP) modules for feature extraction. CSP and SPP mean that the input feature is divided into several branches, and the output features from these branches are fused after different convolutional operations. Thus, the utilization rate of feature information is improved. Next, the feature pyramid strategy is used for integrating the feature maps, which fuse the different layers adequately. In addition, YoloV4++ calculates the total loss based on the focal loss function.

Compared with baseline methods, YoloV4++ has achieved remarkable results. Compared with a two-stage method, namely Faster R-CNN, we increased mAP by 1.52% and FPS by 49.26. In the comparisons with one-stage methods (SSD, YoloV3, YoloV4, and Yolo X), we have achieved the best results on both mAP and FPS. In terms of mAP, the value of YoloV4++ is 9.26% higher than that of SSD, 18.72% higher than that of YoloV3, 2.31% higher than that of YoloV4, and 0.91% higher than that of Yolo X. In terms of FPS, the value of YoloV4++ is 18.41 more than that of SSD, 31.78 more than that of YoloV3, 36.81 more than that of YoloV4, and 15.36 more than that of Yolo X. Experimental results show that YoloV4++ can detect insulator targets well in real time.

To analyze the difference more intuitively, we have drawn their speed versus accuracy diagram, as shown in Figure 11. The closer the point is to the upper right, the better the comprehensive performance of the method is. YoloV4+ and YoloV4++ are on the top right of Figure 11. Faster R-CNN is on the left and YoloV3 is at the bottom. SSD, YoloV4, and Yolo X are in the middle.

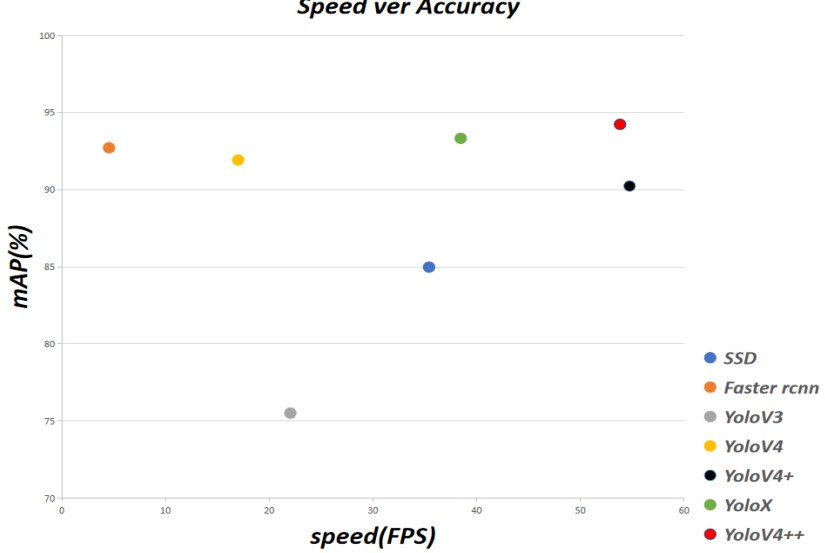

**Figure 11.** Speed (FPS) versus accuracy (mAP) on RSIn-Dataset.

YoloV4+ has the best performance in detection speed while it is below Faster R-CNN, YoloV4, Yolo X, and YoloV4++ in the detection accuracy. This is because YoloV4+ adopts the lightweight backbone network MobileNetV1, which reduces the parameters. However, the lightweight backbone causes inadequate utilization of image information. Thus, YoloV4+ does not have an advantage in detection accuracy.

YoloV4++ is better than SSD, Faster R-CNN, YoloV3, YoloV4, and Yolo X in both detection accuracy and speed. Especially, YoloV4++ has achieved a great improvement in accuracy based on YoloV4, while the detection speed is about three times that of YoloV4. Note that, compared to YoloV4+, YoloV4++ has greater detection accuracy with little decrease in FPS. This is because YoloV4++ employs the focal loss function to raise detection accuracy while using MobileNetV1 to reduce model parameters. Experimental results

show that the proposed lightweight network, YoloV4++, achieves the best balance between insulator detection accuracy and speed.

## 6. Conclusions

In this research, we build RSIn-Dataset to promote the development of deep learning in the power line inspection field. RSIn-Dataset is a power patrol scene dataset for insulator detection. It has distinctive characteristics of high resolution, richness of annotations, extensive backgrounds, and diversity. Due to these advantages, RSIn-Dataset can be used for power device detection tasks. Additionally, we propose the new lightweight network YoloV4++ for detecting insulators. In YoloV4++, MobileNetv1 is used as the backbone to reduce the model parameters. Then, depthwise separable convolution is used in the neck network for a further parameter reduction. To compensate for the decline in accuracy, focal loss is introduced to alleviate the sample imbalance. Moreover, we conduct experiments with mainstream object detection methods and the proposed method on RSIn-Dataset. From the experimental performances, we analyze the advantages and disadvantages of these object detection methods and build a benchmark to supply references for other insulator detection researchers. The experimental results show that RSIn-Dataset can be used for the performance evaluation of object detection and our proposed lightweight network provides greater improvements to the baseline YoloV4 on RSIn-Dataset. In the benchmark, YoloV4++ also achieves the best results of mAP and excellent performance of FPS.

However, there are still some challenges in the detection of power devices in power line inspection. Our method is still inadequate in terms of the detection accuracy and the model parameters could be further reduced. Therefore, our future research will be dedicated to raising the detection accuracy and reducing the model parameters. For example, the method of small object detection or model compression strategies can be employed in the object detection methods. Meanwhile, RSIn-Dataset can be extended by collecting other power device data to apply the dataset to other power device detection tasks.

**Author Contributions:** Funding acquisition, Y.L., T.L. and F.S.; Methodology, Y.L. and S.H.; Software, S.H.; Writing—original draft, S.H.; Writing—review and editing, Y.L., T.L. and F.S. All authors have read and agreed to the published version of the manuscript.

**Funding:** This research was funded by the Guangxi Science and Technology Base and Talent Project (Grant No. Guike AD22080043), the Natural Science Foundation of Guangxi under Grant 2022GXNSFBA035661, the Hubei Key Laboratory of Intelligent Robots (Grant No. HBIR202108), the Research Basic Ability Improvement Project of Young and Middle-aged Teachers in Guangxi Universities (Grant No. 2021KY0015), and Bagui Scholars Project. The APC was funded by Feng Shuang and Yong Li.

**Institutional Review Board Statement:** Not applicable.

**Informed Consent Statement:** Not applicable.

**Data Availability Statement:** Pascal VOC: "http://host.robots.ox.ac.uk/pascal/VOC/ (accessed on 10 November 2022)"; COCO Dataset: "http://mscoco.org/ (accessed on 10 November 2022)"; ImageNet Dataset: "https://image-net.org/ (accessed on 10 November 2022)"; DOTA Dataset: "https://captain-whu.github.io/DOTA/dataset.html (accessed on 10 November 2022)"; Git Dataset: "https://github.com/InsulatorData/InsulatorDataSet (accessed on 10 November 2022)"; RSIn-Dataset: "https://github.com/caigouyihao/Rsin-dataset (accessed on 10 November 2022)".

**Acknowledgments:** The authors thanks Andrew Chen for helping with the draft writing and checking.

**Conflicts of Interest:** The authors declare no conflict of interest.

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
