# Peer review of "RSIn-Dataset: An UAV-Based Insulator Detection Aerial Images Dataset and Benchmark"

_drones, doi:10.3390/drones7020125_

Round 1

Reviewer 1 Report

This paper reports an insulator detection dataset and benchmark, and proposed an improved YoloV4 network for insulator detection. The research topic is hot. The proposed method uses lightweight feature extraction network and employs Focal loss to address the imbalance between samples. The experiments show the excellent performances of the proposed method on the constructed dataset, and demonstrate the superiority over other models. However, there are still some concerns required to be carefully addressed.

1. The description of the introduction is not concise enough, some expressions are not clear enough, and some grammatical representations should be revised.

2. The visual display of experimental results is not clear. The difference between different algorithms cannot be clearly distinguished, and the image scene displayed at present has a high similarity, so it is suggested to increase the comparison of detection effects in different types of scenes

3. The paper has some formatting problems, please check carefully.

4. The authors lack an analysis of the failure or poor performance on the proposed method. It is suggested to add some analysis and explanations for future research directions.

Author Response

Dear reviewer,

Thank you for reading our manuscript carefully. We appreciate the time and effort that you have dedicated to provide your valuable feedback on my manuscript. We have been able to incorporate changes to reflect your advice. As for the specific response, please see the attachment.

Reviewer 2 Report

Dear Authors

Authors of the manuscript raised an important and hot topic in RSIn-Dataset - UAVs studies. There is a high need to estimate and construct a dataset (RSIn-Dataset) for insulator detection in the electric power patrol scene using techniques especially non-destructive techniques. The non-destructive techniques are important to study surface and subsurface processes, as well as better recognition of factors controlling landscapes may lead to develop efficient control measures. Although, I have several doubts concerning this manuscript which in details are given in the attached pdf file.

Author Response

(The authors gave the same response as above.)

Reviewer 3 Report

This research article constructed a novel insulator detection dataset for electric power lines and compared the data sets with several known detection CNN based method. To improve the quality of the paper, the authors need to consider the following comments for the revision.

 Major comments:

1.      It is suggested that the author revises English writing.

2.      In abstract, it is advised to add some major findings from the review and some key drawbacks/scope remain for further study.

3.      In materials and method, please add drone and camera specifications (in table or description). And what is the height or distance the data (image) were acquired. Is it random? If it is, then why? Figure 3 shows only closed view images, are all the dataset images similar? Then what are the effect of the detection methods on closed view insulator images and far distance view images? As one of the objective was to dataset development, author needs to focus on this issue.

4.      Line 180-181, 1887 images and 3286 targets. What is targets? Insulator or angle of view? Also, image resolution ranging from 1152×864 to 7360×4912. Why the resolution changes? Is it for the data mix-up or different resolution of cameras?

5.      Line 179, “we preprocess some images…” What kind of preprocessing was used?

6.      A more detailed descriptions of “Baseline Methods” is recommended.

7.      Section 5.1 should go to the materials and method section (after section 4.2.2). Also Lines 328-342, in section 5, should be described in materials and method (after section 4.2.2).

8.      In results section, need to show some comparison with the similar study with the proposed algorithm.

9.      Is there any issue with the background during the detection? Author did not discuss about this issue.  How background affect/or not in different method?

10.  Conclusion looks very simple. Needed to be written with major finding of this article with drawbacks of the current method. 

Author Response

(The authors gave the same response as above.)

Round 2

Reviewer 2 Report

Dear authors,

The effort to meet the demands requested by the reviewer is notable. The grammar has been modified and improved, meeting the standards required for publication in English. Also, the discussion section are extensively improved. the flowcharts and the references are added well. So, most of my questions were addressed and I have no doubts about the methods used in the study.